# Comparison of the Effect of Phospholipid Extracts from Salmon and Silver Carp Heads on High-Fat-Diet-Induced Metabolic Syndrome in C57BL/6J Mice

**DOI:** 10.3390/md21070409

**Published:** 2023-07-19

**Authors:** Qi Wang, Rui Wang, Xiuju Zhao, Hongyan Lu, Peng Zhang, Xinjie Dong, Yuming Wang

**Affiliations:** 1School of Food Science and Engineering, Wuhan Polytechnic University, Wuhan 430023, China; wss_2019@163.com (R.W.); luhongyan937@whpu.edu.cn (H.L.); zp2022@whpu.edu.cn (P.Z.); 2Key Laboratory for Deep Processing of Major Grain and Oil, Ministry of Education (Wuhan Polytechnic University), Wuhan 430023, China; 3Hubei Key Laboratory for Processing and Transformation of Agricultural Products, Wuhan Polytechnic University, Wuhan 430023, China; 4School of Biology and Pharmaceutical Engineering, Hubei Wuhan Polytechnic University, Wuhan 430023, China; dzrdez@163.com (X.Z.); dxj_stud@163.com (X.D.); 5College of Food Science and Engineering, Ocean University of China, Qingdao 266003, China; wangyuming@ouc.edu.cn

**Keywords:** metabolic syndrome, phospholipid, salmon, silver carp, metabonomic

## Abstract

Metabolic syndrome (MetS) is a global health problem, and EPA/DHA-enriched phospholipids (EPA/DHA-PLs) have been found to have positive effects on MetS improvement. Currently, research on EPA/DHA-PL mainly focuses on special and rare seafood, such as phospholipids derived from krill, sea cucumber, squid, and fish roe. However, it has been recently demonstrated that abundant EPA/DHA-PL can also be found in bulk fish and its by-products. Nonetheless, there is still limited research on the biological activities of EPA/DHA-PL derived from these sources. The aim of this study was to investigate the effect of phospholipid extracts from the heads of salmon and silver carp (S-PLE and SC-PLE) on the high-fat-diet-induced MetS in C57/BL mice. After an 8-week intervention, both SC-PLE and S-PLE had a significant ameliorating effect on MetS. Moreover, SC-PLE was more effective than S-PLE in reducing liver inflammation and fasting glucose. Both of the PL extracts were able to regulate the expression of key genes in lipid synthesis, fatty acid β-oxidation, and insulin signaling pathways. Compared with S-PLE, dietary SC-PLE had a greater influence on liver metabolomics. Pathway enrichment analysis showed that the differential metabolites of SC-PLE were mainly involved in arachidonic acid metabolism and glutathione metabolism. The results indicated that the different metabolic regulation methods of S-PLE and SC-PLE could be related to their variant molecular composition in EPA/DHA-PL.

## 1. Introduction

Metabolic syndrome (MetS) is a group of metabolic disorders that includes insulin resistance, dyslipidaemia, central obesity, and hypertension [1]. In addition to individual genetic factors, the chronic consumption of high-calorie foods and reduced physical activity have contributed to the increased incidence of metabolic syndrome in recent years [2]. Moreover, if left untreated, metabolic syndrome is significantly associated with an increased risk of developing diabetes and cardiovascular diseases (CVDs). The molecular mechanisms underlying metabolic syndrome are not yet fully understood. Among the mechanisms proposed, insulin resistance, chronic inflammation, and neurohormonal activation appear to be key factors in the progression of MetS and its subsequent transition to cardiovascular disease and T2DM [3]. Macrophage infiltration in visceral adipose tissue intervals produces a pro-inflammatory state that promotes insulin resistance [4]. More recently, newer research has assessed the role of various metabolites in the pathogenesis of metabolic-disorder-related diseases [5]. It has reported significant changes in bile salts and glutathione-related biochemicals in patients with NAFLD [6].

n-3 polyunsaturated fatty acid phospholipids (n-3 PUFA-PLs), characterized by n-3 polyunsaturated fatty acids at the sn-2 or sn-1 positions, are positively correlated with the improvement of metabolic abnormalities [7]. EPA (C20:5) and DHA (C22:6) are precursors of anti-inflammatory lipid mediators [8]. A reduction in the amount of n-3PUFA has been observed in metabolic diseases and is speculated to accelerate inflammation. For example, EPA-PC and EPA-PE reduced the levels of TNF-α, IL-6, and MCP-1 in the liver [9]. Dietary EPA-PL reduced macrophage infiltration in the liver and adipose tissues [10]. In addition, n-3 PUFA-PLs in the diet, especially EPA/DHA-PL, can alleviate obesity and insulin resistance [11]. Some kinds of special and rare seafood are important sources of EPA/DHA-PL, such as krill [12], sea cucumber [13], squid, and fish roe [14]. However, most of them are expensive and less productive. Wang et al. showed that the mixture of ethanol and n-hexane (4:1, *v/v*) was a suitable solvent for extracting EPA/DHA-PL from by-products of shrimps [15]. Currently, there is a growing focus on the isolation and preparation of EPA/DHA-PL from by-products of seafood processing. Therefore, future research will primarily concentrate on investigating their biological activities.

The existence of small molecule metabolites in the body is closely related to the occurrence and development of diseases [16]. Metabolites in biological samples can be qualitatively and quantitatively analyzed by metabonomic analysis, so as to find the biomarkers of diseases [17]. Extensive targeted metabolomics has the advantages of both non-targeted and targeted metabolomics [18]. It has better repeatability and higher metabolic coverage. In recent years, a large number of studies have shown that the development of MetS may be related to liver metabolites [19]. However, there are few studies on the effects of phospholipids from aquatic products on liver metabolites.

In the face of the increasing demand for dietary supplements of EPA/DHA-PL, the use of fish-processing by-products has become crucial. Salmon is a promising marine biological resource found in the high latitude waters of Norway, the USA, and Canada [20]. Silver carp has the second-largest production amount of all freshwater farmed fish in China and has great potential for resource exploitation [21]. The heads of salmon and silver carp are by-products of processing and have been detected to be rich in n-3 PUFA PLs [22]. In this study, we aimed to explore the effect of salmon phospholipid extracts (S-PLE) and silver carp phospholipid extracts (SC-PLE) on high-fat-diet-induced MetS mice. In addition, metabolomic analysis was performed in order to explore the underlying mechanisms. Moreover, the potential role of lipid synthesis, fatty acid β-oxidation, and the IRS/PI3K/AKT signaling pathway were also assessed.

## 2. Results

### 2.1. Phospholipid Composition of SC-PLE and S-PLE

The content and number of phospholipid molecules in the two fish species is shown in Table 1. The phospholipid contents of S-PLE and SC-PLE were 65.49% and 66.16%, respectively. PC and PE, the most abundant PLs in the two species, played vital roles in the growth and metabolism of the organisms. Lower levels of LPC, LPE, and PI were detected in the two species; however, these lipids are vital to the fluidity of cellular membranes and signaling process. As one of the most important features of PLs, high levels of PUFAs were detected in the fatty acid chain of PL, especially in DHA and EPA. In this study, high percentages of DHA and EPA were present in the fatty acid chain of PC and PE (Table 2). About 4.29–5.49% of DHA-PC, 2.28–4.95% of EPA-PC, 1.07–4.98% of DHA-PE, and 0.16–0.58% of EPA-PE were detected in the two species. DHA-PC accounted for a large proportion of PUFA-PL and was the highest in S-PLE (5.49% of total PL), while EPA-PC was the highest (4.95% of total PL) in SC-PLE.

### 2.2. The Effect of S-PLE or SC-PLE on HFD-Induced Obesity and Hyperlipidemia

As shown in Table 3, HFD led to significant obesity and hyperlipidemia. Compared to the HF group, dietary S-PLE and SC-PLE significantly suppressed body weight gain, epididymal white adipose tissue (WAT), and serum levels of TG and TC (*p* < 0.05). In the S-PLE group, the body weight gain and serum levels of TG and TC decreased by 13.7%, 14.3%, and 16.7%, respectively (*p* < 0.05). In the SC-PLE group, the levels of body weight and serum lipid decreased by 53.44%, 36.1%, and 33.6%, respectively (*p* < 0.05), for body weight gain, serum TG, and serum TC. The results showed that the degree of decrease in the SC-PLE group was greater than that in the S-PLE group, indicating that SC-PLE had a greater improvement effect on obesity and hyperlipidemia.

### 2.3. The Effect of S-PLE or SC-PLE on HFD-Induced Glucose Tolerance and Insulin Sensitivity

To evaluate the effect of S-PLE or SC-PLE on glucose tolerance, an oral glucose tolerance test (OGTT) was performed. The results are shown in Figure 1. The glucose tolerance of the HFD group was impaired compared to that of the ND group. The concentration of serum glucose first increased and then decreased, and it reached a peak at 30 min in all of the groups. However, the level of serum glucose in the SC-PLE group decreased at 30 min compared with that of the HFD group, which indicated that SC-PLE restrained the rise in serum glucose levels. Nevertheless, both SC-PLE and S-PLE did not affect the AUC in the HFD-fed mice (Figure 1). The serum levels of fasting glucose and insulin were measured, as shown in Figure 1. Dietary SC-PLE supplementation notably reduced the elevated serum levels of fasting glucose and insulin (*p* < 0.05) in the HFD mice. However, S-PLE had no significant influence on the serum levels of fasting glucose and insulin. In a word, the SC-PLE group showed a greater improvement in glucose tolerance and insulin sensitivity than the S-PLE group.

### 2.4. The Effect of S-PLE or SC-PLE on HFD-Induced Liver Fat Accumulation

The histopathological changes in the liver tissue of the mice were observed with a fluorescence microscope (Figure 2). The liver tissue slice of the ND group represented the normal liver state, which showed that the liver lobules were clear and complete. In contrast, the HFD group showed that the lipid droplets accumulated in the cytoplasm and nuclei at the edges of the liver cells, indicating the presence of lipids in the liver. Both the S-PLE and the SC-PLE treatments significantly improved the cellular structure of the liver tissue and reduced lipid droplets in HFD-induced mice compared to the HFD group.

The mice in the HFD group developed liver fat accumulation, and the levels of TG and TC in the liver were significantly higher than those in the ND group (*p* < 0.05) (Table 4). Moreover, S-PLE and SC-PLE improved the liver lipid levels to different degrees. Compared with the HFD group, the SC-PLE treatment consistently reduced the levels of TG and TC in the livers of the mice significantly (*p* < 0.05). The S-PLE treatment significantly decreased the levels of TG in the livers of the mice compared with the HFD group (*p* < 0.05) but had no significant influence on the levels of hepatic TC in the S-PLE group (*p* > 0.05).

### 2.5. The Effect of S-PLE or SC-PLE on HFD-Induced Chronic Liver Inflammation

To evaluate the effect of SC-PLE and S-PLE on chronic inflammation, levels of inflammatory cytokines in the liver were measured. The results are shown in Figure 3. The levels of TNF-α, IL-6, MCP-1, and IL-1β in the HFD group were significantly higher than those in the ND group (*p* < 0.05). Dietary SC-PLE supplementation reduced the levels of TNF-α, IL-6, IL-1β, and MCP-1 (*p* < 0.05) in the liver compared to the HFD group. However, dietary S-PLE administration had no influence on levels of inflammatory cytokines in the liver. The results indicated that the SC-PLE group had better efficacy in the intervention of chronic inflammation.

### 2.6. Effect of S-PLE and SC-PLE Supplementation on mRNA Expressions Associated with Lipid Metabolism in the Liver

The effect of S-PLE and SC-PLE on lipid metabolism was investigated by using q-PCR analysis in the liver, which is the largest visceral organ responsible for maintaining lipid homeostasis. As demonstrated in Figure 4, compared with the ND group, the relative mRNA expression levels of G6PDH and FAS in the HFD group significantly increased (*p* < 0.05). In contrast, the relative mRNA expression level of PPAR-α reduced considerably (Figure 5A). Compared with the HFD group, the relative expression level of these genes was highly reversed in the SC-PLE group and the S-PLE group. These results demonstrated that both the SC-PLE treatment and the S-PLE treatment could improve the expression of lipid metabolism genes.

### 2.7. Effect of S-PLE and SC-PLE Supplementation on mRNA Expressions Associated with the IRS/PI3K/Akt Signaling Pathway in Liver

The IRS/PI3K/AKT signaling pathway is the primary method for insulin to mediate glucose metabolism in the liver. As demonstrated in Figure 6, compared with the ND group, the relative mRNA expression levels of IRS1, Akt1, and GLUT2 in the liver tissues from the HFD group significantly decreased (*p* < 0.05). Compared with the HFD group, the relative mRNA expression level of IRS1 and Akt1 were reversed in the SC-PLE and S-PLE groups. In addition, compared with the HFD group, the mRNA expression level of GLUT2 in the S-PLE group significantly increased (*p* < 0.05), while the mRNA expression level of GLUT2 in the S-PLE group remained unchanged compared with that of the high-fat group.

### 2.8. Liver Metabonomic Profile Analysis

To further investigate the regulatory effect of SC-PLE and S-PLE on glycolipid metabolism, non-targeted metabolomics analysis of the mouse livers was performed. A total of 993 metabolites, including 339 amino acids and metabolites, 132 fatty acids, 130 organic acids and derivatives, and 83 nucleotides and metabolites, were detected in the liver samples. Orthogonal partial least squares discriminant analysis (OPLS-DA) is the most commonly used data analysis method in metabolomics and was selected for the dimension reduction analysis of metabolites in this study. The OPLS-DA score map for the different groups is illustrated in Figure 7. The degree of aggregation within the groups was good and the separation among the four different groups was complete. Subsequently, differential fold-change values (>2 or <0.5) and variable-important-in-projection (VIP) values (>1) for metabolites were used to screen for differential metabolites. The red dots in the volcanic map represent up-regulated levels of metabolites and the green dots represent down-regulated levels of metabolites (Figure 8). The levels of 82 metabolites were different in the ND vs. the HFD groups. The values for the differentially expressed metabolites in the HFD vs. S-PLE groups and the HFD vs. SC-PLE groups were 43 and 128, respectively.

Finally, we identified 12, 9, and 10 potential differential metabolites in ND vs. HFD, HFD vs. S-PLE, and HFD vs. SC-PLE, respectively. As shown in Table 5, compared with ND, there were eight metabolites up-regulated in HFD, including L-cystine, thromboxane B2, taurocholic acid, 13-oxo-9Z, 11E-octadecadienoic acid, 9-oxo-10E, 12Z-octadecadienoic acid, and 9,12,13-trihydroxy-octadecadienoic acid. At the same time, seven metabolites were down-regulated, including γ-L-glutamyl-L-cysteine, reduced glutathione, cortisol, taurocholic acid, 11-cis-retinol, all-trans-retinal, and cysteinylglycine. Compared with the HFD group, four substances were up-regulated in the S-PLE group, including N,N′-diacetylchitosan, deoxyguanosine diphosphate, 5′-adenosine sulfate, and (±)17-HDHA, and five metabolites were down-regulated, including carnosine, 4-hydroxy-2-ketoglutarate, DL-3,4-dihydroxy amygdalic acid, 13-hydroxy-9Z,11E,15Z-octadecatrienoic acid, and carnitine C8:0 (Table 6). In the liver, compared with HFD, the levels of glutathione, gamma-L-glutamyl-L-cysteine, cysteinylglycine, and (±) 5-HEPE were significantly increased by S-PLE, whereas the levels of thromboxane B2, prostaglandin A2, 15-keto-prostaglandin F2a, 11-beta-prostaglandin PGF2a, 12-prostaglandin J2, and 3,5-dihydroxy-3-methylpentanoic acid were decreased (Table 7).

The pathways of biomarkers among the groups were analyzed with KEGG, and four related metabolic pathways were identified (Figure 9). The high-fat diet interfered with multiple hepatic metabolic pathways, and the main metabolic pathways based on biomarkers were glutathione metabolism, retinol metabolism, bile acid metabolism, and linoleic acid metabolism (*p* < 0.05). SC-PLE intervention regulated arachidonic acid metabolism and glutathione metabolism, while the differential metabolites produced by S-PLE intervention were not enriched into significantly regulated pathways in the liver.

## 3. Discussion

The principal feature of MetS is that various metabolic disorders coexist in the same body, which can be regarded as a manifestation of a long-term high-calorie diet [23]. The disease components of MetS mainly include insulin resistance, hyperinsulinemia, abnormal blood sugar levels, dyslipidemia, and hypertension [2]. Our study demonstrates that S-PLE and SC-PLE have the potential to ameliorate obesity, dyslipidemia, hepatic lipid accumulation, hepatic inflammation, and insulin resistance, thereby aiding in the prevention of MetS caused by a high-fat diet.

Numerous studies have indicated that consuming an appropriate amount of PUFA (such as EPA and DHA) can effectively reduce blood lipid levels and the incidence of cardiovascular diseases [24,25,26]. S-PLE and SC-PLE are rich in EPA/DHA-PL, which may be the primary reason for them improving MetS. n-3 PUFA serve as natural ligands for peroxisome proliferator-activated receptor α (PPAR-α) and modulate the expression of sterol regulatory element-binding protein (SREBP) [27]. The SREBP-1c regulates the transcriptional expression of key enzymes in the lipogenesis pathway, such as acetyl-CoA carboxylase (ACC), fatty acid synthase (FAS), and stearoyl-CoA desaturase (SCD) [28]. The activation of PPAR-α, on the other hand, effectively increases the expression of peroxisome proliferator-activated receptors and enzymes associated with fatty acid beta-oxidation, such as acyl-CoA oxidase (ACO) and carnitine palmitoyltransferase (CPT) [29,30]. In fact, gene expression analysis revealed that S-PLE and SC-PLE can inhibit the expression of FAS and G6PDH, while activating PPAR-α and its downstream genes CPT-1a and CPT-2. This suggests that S-PLE and SC-PLE could regulate hepatic lipid metabolism disruption by modulating both lipid synthesis and fatty acid beta-oxidation. Ding et al. investigated the effects of DHA-PC and DHA-PS on age-related lipid metabolism disorders in SAMP8 mice [31]. The results demonstrated that both dietary DHA-PC and DHA-PS significantly reduced serum and hepatic lipid levels by inhibiting SREBP-1c-mediated fatty acid synthesis in the liver [31]. Furthermore, they activated hepatic fatty acid oxidation mediated by PPAR-α in SAMP8 mice. Through protein–lipid overlay experiments, Tian et al. demonstrated that EPA-PC and EPA-PE isolated from sea cucumber can bind to PPAR α/PPAR γ [32]. Additionally, they validated through preadipocyte differentiation experiments that EPA-PC and EPA-PE improve lipid accumulation and insulin resistance in mice by activating PPAR α/γ [32]. The experimental results from these studies are consistent with our findings.

The regulation of liver insulin metabolism is mainly mediated by the IRS/PI3K/AKT pathway [33]. The dysregulation of the IRS/PI3K/Akt signaling pathway is a direct triggering factor for insulin resistance [33]. IRS serves as a crucial substrate in insulin signaling transduction [34]. A high-fat diet suppresses the gene expression of IRS1, leading to the inhibition of PI3K and AKT activation [35]. Subsequently, the expression of downstream genes associated with glucose transport, such as GLUT2, is down-regulated. Ultimately, this results in a reduced ability of the body to utilize glucose, contributing to insulin resistance [36]. In this study, it was observed that S-PLE and SC-PLE up-regulated the expression of IRS1 and Akt1 in the IRS/PI3K/AKT pathway of high-fat-diet-fed mice. This indicates that S-PLE and SC-PLE can inhibit insulin resistance by mediating the IRS/PI3K/AKT pathway. There are various mechanisms involved in the disruption of the IRS/PI3K/Akt signaling pathway. Among them, chronic inflammation in adipose tissue plays a crucial role [37]. In vivo studies have demonstrated that TNF-α, IL-6, and IL-1β can directly impair the insulin signaling pathway by increasing the serine phosphorylation of IRS [37]. The research conducted by Gao et al. showed that EPA/DHA-PC could inhibit the activation of the NF-κB and JNK inflammatory pathways induced by a high-fat diet [11]. This leads to a reduction in the release of pro-inflammatory cytokines, thereby suppressing insulin resistance [11]. Our study also found that S-PLE and SC-PLE can reduce the levels of inflammatory factors in the liver. S-PLE and SC-PLE are rich in EPA/DHA-PC, which might be one of the reasons for their improvement of insulin resistance.

In addition to their ability to modulate the signaling pathways related to hepatic lipid metabolism and insulin resistance, S-PLE and SC-PLE exhibit significant effects on liver metabolism. Through potential differential metabolite screening and KEGG pathway analysis, it was found that these effects manifest as alterations in the metabolites involved in arachidonic acid metabolism and glutathione metabolism, as well as an increase in lipid metabolites associated with PUFA. Under the inflammatory stimulation induced by a high-fat diet, the activation of phospholipase A2 leads to the release of arachidonic acid (ARA) from membrane phospholipids at the sn-2 position [38]. ARA is subsequently converted by several enzymes, including cyclooxygenases (COX), cytochrome P450 enzymes, and lipoxygenases, into a variety of eicosanoids, such as prostaglandins and thromboxanes [39]. Therefore, extensive research indicates a positive correlation between the metabolites of arachidonic acid and the development of inflammation [40,41,42]. Our study demonstrates that intervention with SC-PLE results in the downregulation of the metabolites associated with arachidonic acid metabolism, including thromboxane B2, prostaglandin A2, 15-keto-prostaglandin F2a, 11-beta-prostaglandin PGF2a, and 12-prostaglandin J2. Simultaneously, an upregulation of oxidized lipid metabolites of EPA/DHA was observed in the intervention with S-PLE and SC-PLE. Both n-3 PUFA and arachidonic acid serve as substrates for COX-1 [43]. Therefore, consuming lipid sources rich in n-3 PUFA can exert a positive impact on inflammatory diseases through the action of COX-1. Furthermore, a high-fat diet can lead to hepatic oxidative stress, which is manifested by the downregulation of reduced glutathione in metabolites. Intervention with SC-PLE upregulates the metabolites of glutathione, indicating an alleviation of hepatic oxidative status. Sun et al. demonstrated that Antarctic krill oil can inhibit the hepatic oxidative damage induced by a high-fat diet, while reducing malondialdehyde (MDA) levels and increasing superoxide dismutase (SOD) levels [44]. Migliaccio et al. also provided evidence that a high-fat diet enriched with fish oil (HFO diet, primarily n-3 PUFA) can improve mitochondrial function and reduce the production of reactive oxygen species in the liver and skeletal muscle [45]. Therefore, we speculate that SC-PLE, which is rich in EPA/DHA-PL, may have a protective effect against hepatic oxidative stress.

In terms of the improvement of MetS, the effects of SC-PLE surpass those of S-PLE. Specifically, SC-PLE demonstrates a superior regulation of glucose intolerance, hepatic inflammation, and hepatic lipid metabolism abnormalities compared to S-PLE. This difference in efficacy may be attributed to the distinct phospholipid compositions of S-PLE and SC-PLE. While both S-PLE and SC-PLE are primarily composed of DHA phospholipids, SC-PLE exhibits higher levels of EPA phospholipids, particularly EPA-PC, while the proportion of DHA phospholipids in S-PLE is slightly higher than that in SC-PLE. A study focusing on atherosclerosis demonstrated that EPA-PL exhibited superior improvement in atherosclerotic lesions induced by a high-fat diet in apoE-/- mice compared to DHA-PL [46]. EPA-PL, but not DHA-PL, significantly reduced serum and hepatic lipid levels by modulating the mRNA and protein levels of genes related to hepatic cholesterol metabolism [46]. Therefore, we speculate that the difference in EPA/DHA phospholipid composition is the reason for the differential effects of S-PLE and SC-PLE in improving MetS. However, further analysis is needed in order to explore the potential mechanisms underlying the distinct effects of DHA-PL and EPA-PL in the context of MetS.

## 4. Materials and Methods

### 4.1. The Preparation and Characterization of SC-PLE and S-PLE

Using the method reported by Wang et al. [15], phospholipid crude extracts were extracted from silver carp heads and salmon heads with 80% (*v*/*v*) ethanol solution. Briefly, 10 g of freezing dried powder was completely mixed with 80 mL of 80% (*v*/*v*) ethanol solution. The mixture was then stirred quickly and allowed to stand at 35 °C for 4 h, and the separation was performed by suction filtration. Then, the liquid was collected, and the solvent was removed using vacuum rotary evaporation at 65 °C. Finally, lipid compositions were extracted with chloroform–methanol solvent (2:1, *v*/*v*) from the phospholipid crude extract and were washed with acetone. The SC-PLE and S-PLE were collected in small shading brown bottles and stored at −18 °C.

Both SC-PLE and S-PLE were separated with Shimadzu UPLC LC-30A ultra-performance liquid chromatography and analyzed via mass spectrometry with a Triple TOF mass spectrometer (UPLC-Q-TOFMS, Triple TOF 6600 system, AB Sciex, Concord, ON, Canada). The phospholipid molecular species of SC-PLE and S-PLE were determined by using the previous method and the results are shown in Appendix A [47].

### 4.2. Animal Grouping and Administration

All animal experiments were approved by the Animal Ethics Committee of Experimental Animal Care at the College of Food Science and Engineering, Ocean University of China (Qingdao, China; Approval no. SPXY2020101401). Five-week-old male C57BL/6J mice were purchased from the Vital River Laboratory Animal Center (Beijing, China). The mice were housed under specific pathogen-free conditions at a constant humidity of 65 ± 15% and a temperature of 24 ± 2 °C. After one week of adaption, the animals were randomly assigned to one of the following four groups (*n* = 10 per group): normal group (ND), high-fat-diet group (HFD), SC-PLE group, or S-PLE group. The normal group was fed with a normal chow diet. The remaining three groups were fed with a high-fat diet modified based on AIN-93G. Both feeds were from Research Diets, Inc. (New Brunswick, NY, USA). The HFD contents were 45% Kcal from fat and the ND contents were 10% Kcal from fat. The mice in the HFD group and ND group were orally gavaged with 1.2 g/kg BW of normal saline daily. The S-PLE group and SC-PLE group were orally gavaged with 1.2 g/kg BW of S-PLE emulsion and SC-PLE emulsion, respectively, on a daily basis. The gavage emulsions were obtained by dissolving S-PLE and SC-PLE in saline. The gavage dose was calculated based on the content of n-3PUFA in SC-PLE and S-PLE. Table 2 shows the fatty acid composition of SC-PLE and S-PLE. The previous literature suggested that n-3 PUFA supplementation in animal models of high-fat diet is mg/kg BW, which is approximately equivalent to the 1.2 g/kg daily gavage emulsion required for the mice in this study. After 9 weeks of treatment, the mice were sacrificed following a 12 h fasting period. Blood was collected to separate serum. Fresh tissue samples were fixed for histopathology determination or quick-frozen with liquid nitrogen.

### 4.3. Oral Glucose Tolerance Test (OGTT)

After 7 weeks of feeding, an oral glucose tolerance test (OGTT) was performed on the mice in each group. The OGTT was performed by detecting the blood glucose levels at 0, 0.5, 1, and 2 h after oral gavage of 2 g kg^−1^ glucose to the 12-h-fasted mice. The glucose levels were measured with an ACCU-CHEK Performa glucometer (Roche, USA). The area under the curve of the OGTT was calculated using the following equation: AUC = 0.25 × A + 0.5 × B + 0.75 × C + 0.5 × D, where A, B, C, and D represent the blood glucose levels at 0, 0.5, 1, and 2 h, respectively.

### 4.4. Serum and Liver Biochemical Measurements

The serum and liver concentrations of total cholesterol (TC), triglyceride (TG), and high-density lipoprotein cholesterol (HDL-C) were measured with commercial kits (Najing Jiancheng Bioengineering Institute, Nanjing, China). The hepatic levels of TNF-α, IL-6, MCP-1, and IL-1β were measured using ELISA kits (Beijing Suo Laibao Biotechnology, Beijing, China). The serum levels of insulin were accessed with ELISA kits (Cloud-Clone Company, Wuhan, China).

### 4.5. Histological Analysis

The liver tissues were removed rapidly from the mice and subsequently fixed in 4% formalin, paraffin embedded, sectioned, and, finally, stained with hematoxylin and eosin (H&E). The microscopic structure of the liver was observed and photographed using a fluorescence microscope (Eclipse Ci, Nikon, Tokyo, Japan).

### 4.6. Real-Time PCR Assay

The total RNA of the liver tissue was extracted with RNAex (Accurate Biology, Changsha, China). Real-time qPCR was performed according to the method described previously [48]. The primers were synthesized by Shanghai Sangon Gene Company (Wuhan, China) and the sequences are listed in Appendix A. The gene expression level was calculated using the comparative 2^−∆∆Ct^ method and normalized to that of GAPDH, which was used as an internal control.

### 4.7. Metabolic Profiling Platform

The livers were thawed on ice, after which 400 μL of ice-cold 70% methanol (*v*/*v*) was added to 20 mg of liver. The mixture was swirled for 5 min and then centrifuged at 12,000 rpm and 4 °C for 10 min. The resulting supernatant was collected and centrifuged at 12,000 rpm and 4 °C for 3 min. Then, 200-μL aliquots of supernatant were transferred for LC-MS analysis.

A UPLC system (), with a Waters ACQUITY UPLC HSS T3 C18 column (1.7 µm × 2.1 mm × 100 mm) coupled to a QTRAP mass spectrometer (UPLC and MS/MS: Applied Biosystems 4500 Q TRAP, Sciex, Foster City, CA, USA), was utilized to carry out the liver metabolic profiling. The mobile phase consisted of ultra-pure water (A) and acetonitrile (containing 0.1% formic acid) (B). The elution gradient was set as follows: 0 min, 0% B; 0–11.0 min, 90% B; 11.0–12.0 min, 90% B; 12.0–12.1 min, 5% B; and 10.1–14.0 min, 5% B. The flow rate was 0.4 mL/min, and the injected volume was 2 μL. The electrospray ionization (ESI) source operation parameters were as follows: source temperature, 500 °C; ion spray voltage (IS), 5500 V (positive), −4500 V (negative); ion source gas I (GSI), gas II (GSII), and curtain gas (CUR) were set at 55, 60, and 25.0 psi, respectively; and the collision gas (CAD) was high. On the basis of the multiple reaction monitoring (MRM) mode, Analyst 1.6.3 (Sciex) was applied to acquire MS/MS spectra.

The original data of MS were converted to the mzXML format using Proteo Wizard and untargeted peak detection, peak extraction, peak alignment, and peak integration. The metabolites were identified using the in-house MS2 database (Metware).

After preprocessing the original data, OPLS-DA was applied to process the metabolomics analysis using R software (version 1.0.1). The fitness and reliability of the OPLS-DA were verified by the parameter values R2 and Q2. The permutation test of OPLS-DA was used to validate the model. The metabolites with significant differences (*p* < 0.05) were searched in the Kyoto Encyclopedia of Genes and Genomes (KEGG) database (www.genome.jp/kegg/ accessed on 23 September 2020).

### 4.8. Statistical Analysis

All data were expressed as Mean ± SD. The statistics were performed with Origin 9.0 software (Origin Lab, Northampton, MA, USA) and GraphPad Prism 7.0 software (La Jolla, CA, USA). One-way analysis of variance (ANOVA) and Tukey’s multiple tests were used to determine the significant differences in mean among the diet groups (*p* < 0.05)

## 5. Conclusions

In summary, both SC-PLE and S-PLE exhibit beneficial effects in improving MetS induced by a high-fat diet. SC-PLE and S-PLE alleviate obesity in mice and have preventive effects on hyperlipidemia, hyperglycemia, and hepatic lipid accumulation. Additionally, SC-PLE and S-PLE reduce the levels of hepatic inflammatory factors, namely TNF-alpha, IL-6, IL-1beta, and MCP-1. Furthermore, we observed that SC-PLE and S-PLE can improve hepatic lipid metabolism abnormalities and insulin resistance by modulating lipid synthesis, fatty acid beta-oxidation, and the IRS/PI3K signaling pathway. This study can provide reference for developing new functional lipid by-products of aquatic product processing and preventing MetS.

## Figures and Tables

**Figure 1 marinedrugs-21-00409-f001:**
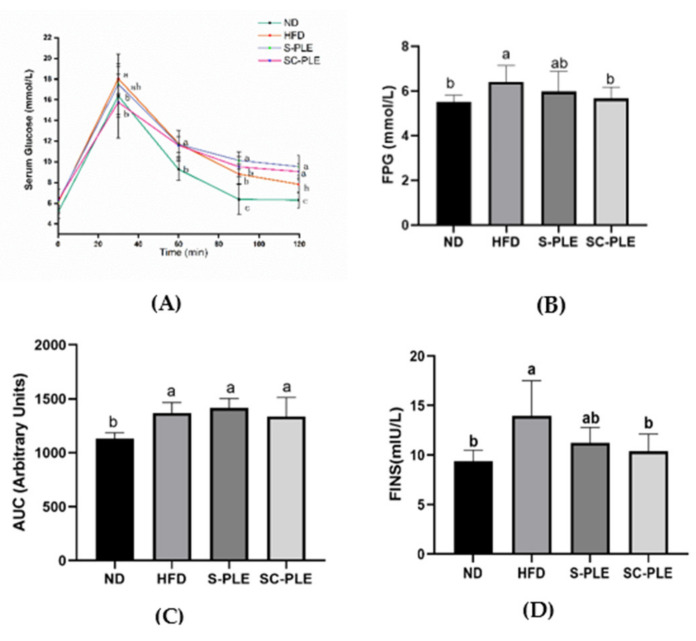
Effect of S-PLE and SC-PLE on glucose homeostasis. The different lowercase letters indicated significant differences in the four groups (*p* < 0.05). (**A**) Serum Glucose; (**B**) Fasting Plasma Glucose (FPG); (**C**) Area Under Curve of Blood Glucose (AUC); and (**D**) Fasting Insulin (FINS).

**Figure 2 marinedrugs-21-00409-f002:**
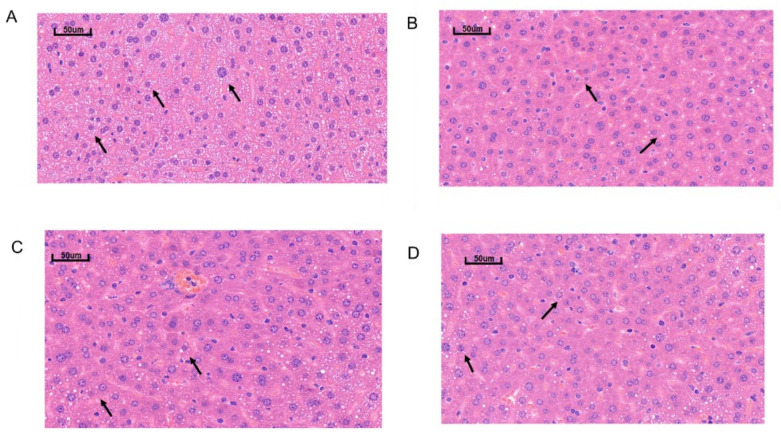
Effect of S-PL and SC-PL on structure changes in liver tissue in MetS mice. (**A**) HFD group; (**B**) ND group; (**C**) S-PLE group; and (**D**) SC-PLE group. The lipid droplets accumulated in the cytoplasm and nuclei (arrows).

**Figure 3 marinedrugs-21-00409-f003:**
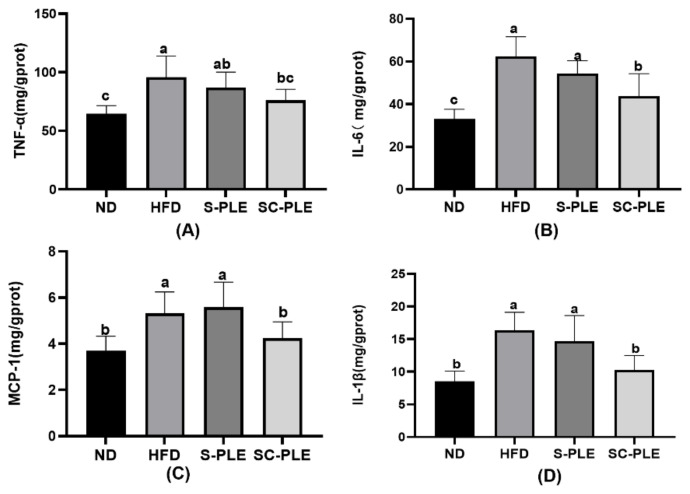
Effect of S-PLE and SC-PLE on liver inflammatory cytokines in MetS mice. (**A**) TNF-α; (**B**) IL-6; (**C**) MCP-1; and (**D**) IL-1β. The different lowercase letters indicate significant differences in the four groups (*p* < 0.05).

**Figure 4 marinedrugs-21-00409-f004:**
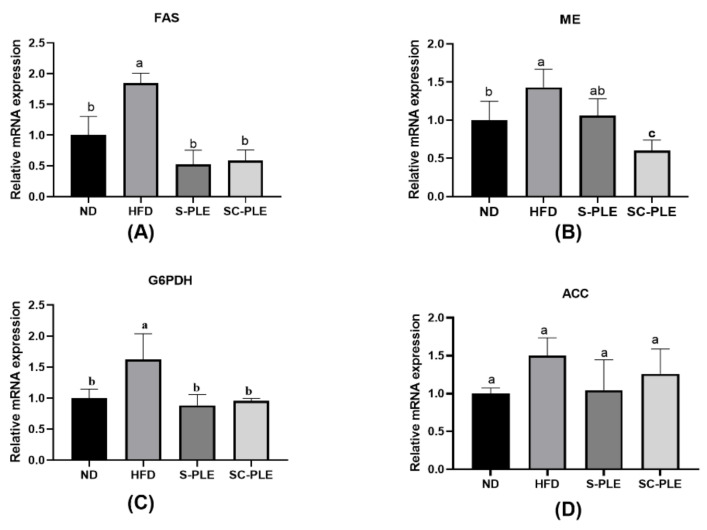
Effect of S-PLE and SC-PLE on the expression of genes related to lipid anabolism in liver tissue. (**A**) Fatty Acid Synthetase (FAS); (**B**) Malic Enzyme (ME); (**C**) Glucose-6-Phosphate Dehydrogenase (G6PDH); **(D**) Acetyl-CoA Carboxylase (ACC). The different lowercase letters indicate significant differences in the four groups (*p* < 0.05).

**Figure 5 marinedrugs-21-00409-f005:**
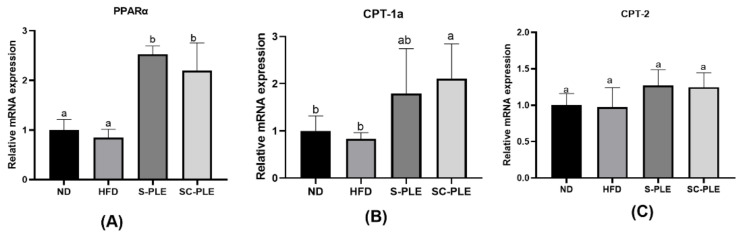
Effect of S-PLE and SC-PLE on the expression of genes related to fatty acid β-oxidation in liver tissue. (**A**) Peroxisome Proliferator Activated Receptor α (PPAR-α); (**B**) Carnitine Palmitoyltransferase 1A (CPT-1A); (**C**) Carnitine Palmitoyltransferase-2 (CPT-2). The different lowercase letters indicate significant differences in the four groups (*p* < 0.05).

**Figure 6 marinedrugs-21-00409-f006:**
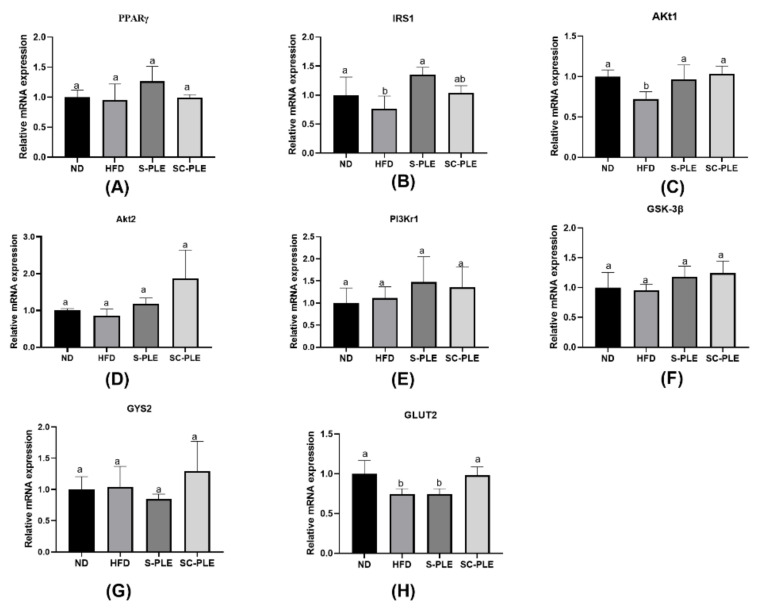
Effect of S-PLE and SC-PLE on the expression of genes related to the IRS/PI3K/Akt signaling pathway in liver tissue. (**A**) Peroxisome Proliferator Activated Receptor γ (RRARγ); (**B**) Insulin receptor substrate 1 (IRS1); (**C**) Protein Kinase B1 (Akt1); (**D**) Protein Kinase B2 (Akt2); (**E**) PI 3 Kinase p85 alpha (PI3Kr1); (**F**) Glycogen Synthase Kinase-3β (GSK-3β); (**G**) Glycogen Synthase 2 (GYS2); (**H**) Recombinant Glucose Transporter 2 (GLUT 2). The different lowercase letters indicate significant differences in the four groups (*p* < 0.05).

**Figure 7 marinedrugs-21-00409-f007:**
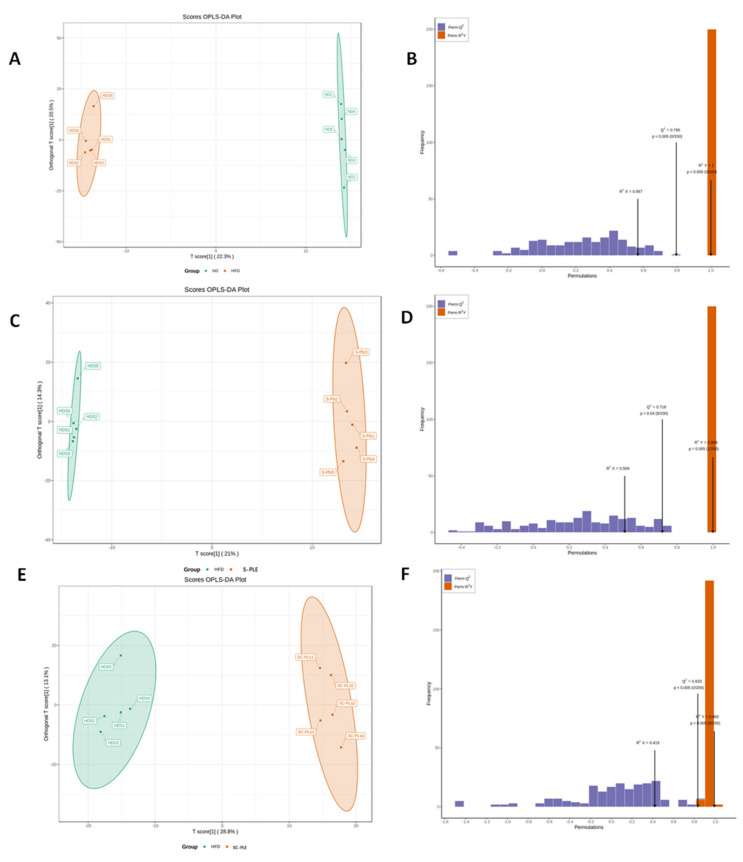
Orthogonal partial least squares discriminant analysis (OPLS−DA) score plots and permutation test of the OPLS−DA model. (**A**,**B**): ND vs. HFD; (**C**,**D**): HFD vs. S−PLE; and (**E**,**F**): HFD vs. SC−PLE.

**Figure 8 marinedrugs-21-00409-f008:**
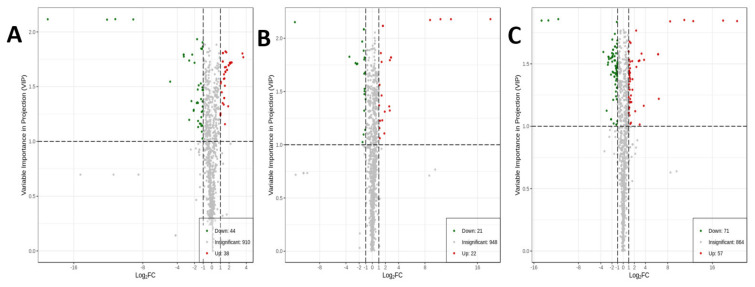
Variable−important-in-projection (VIP) plots in OPLS-DA analysis of the serum metabolites. (**A**) ND vs. HFD; (**B**) HFD vs. S-PLE; and (**C**) HFD vs. SC-PLE.

**Figure 9 marinedrugs-21-00409-f009:**
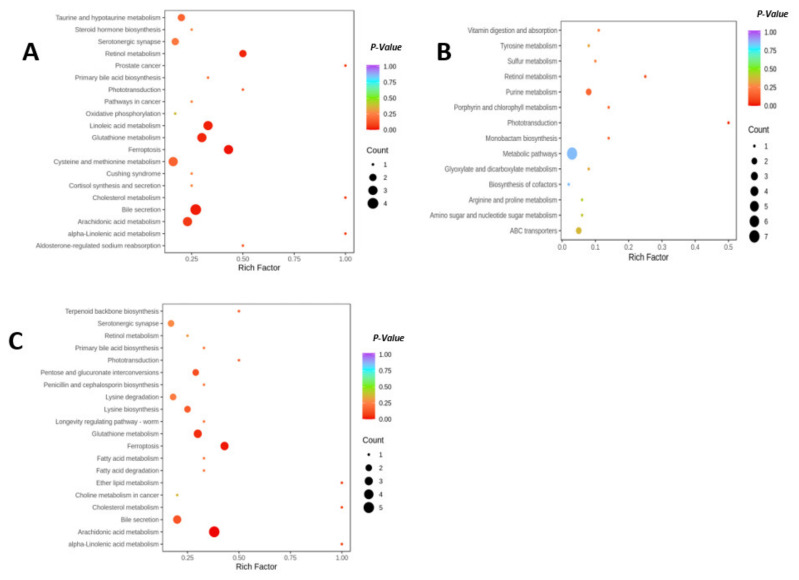
Kyoto Encyclopedia of Genes and Genomes (KEGG) enrichment plots of different metabolites. (**A**) ND vs. HFD; (**B**) HFD vs. S-PLE; and (**C**) HFD vs. SC-PLE. The bubble size is proportional to the impact of each pathway and bubble color denotes the significance from highest in red to lowest in purple.

**Table 1 marinedrugs-21-00409-t001:** Number and content of phospholipid molecules in phospholipid extracts of salmon and silver carp heads (%, mol/mol).

Lipid Classes	S-PLE		SC-PLE	
	Number of Species	Content/%	Number of Species	Content/%
PE	108	31.16	119	15.72
PC	46	43.38	55	56.53
LPC	15	12.37	23	15.15
LPE	20	2.75	19	0.93
PG	41	1.08	20	0.15
PS	16	1.56	16	1.04
LPA	3	0.06	5	0.72
LPG	13	1.22	7	0.11
LPI	3	0.06	2	0.01
LPS	9	1.05	11	1.04
PA	6	0.27	23	2.32
PI	14	5.40	19	6.25

**Table 2 marinedrugs-21-00409-t002:** Characteristic of the composition of EPA/DHA-PL in phospholipid extracts of salmon and silver carp heads (%, mol/mol).

Lipid Classes	PUFA Species	S-PLE/%	SC-PLE/%
PC	EPA	2.28	4.95
	DHA	5.49	4.29
PE	EPA	0.58	0.16
	DHA	4.98	1.07
PE-O	EPA	0.38	0.13
	DHA	1.35	1.38
PE-P	EPA	0.27	0.17
	DHA	1.68	0.86
LPC	EPA	0.36	1.02
	DHA	0.69	6.22
LPE	EPA	0.06	0.02
	DHA	0.46	0.30
PG	EPA	0.01	0.01
	DHA	0.20	0.04
PS	EPA	/	0.01
	DHA	0.14	0.14
PI	EPA	0.86	1.01
	DHA	1.06	0.18
Total PL	EPA	4.79	7.48
	DHA	16.05	14.47

**Table 3 marinedrugs-21-00409-t003:** Effects of S-PLE and SC-PLE on growth and serum lipids in HFD mice.

	ND	HFD	S-PLE	SC-PLE
Growth Parameters				
Food intake (g/day)	2.83 ± 0.50 ^a^	2.57 ± 0.50 ^b^	2.37 ± 0.38 ^b^	2.45 ± 0.46 ^b^
Caloric intake (kcal/day)	10.91 ± 1.91 ^a^	12.16 ± 2.38 ^ab^	11.21 ± 1.82 ^a^	11.58 ± 2.19 ^a^
Initial BW (g)	20.75 ± 0.74	22.00 ± 0.85	21.40 ± 0.70	21.76 ± 0.79
Final BW (g)	26.00 ± 1.64 ^bc^	29.35 ± 1.90 ^a^	26.81 ± 0.90 ^b^	25.32 ± 1.22 ^c^
BW gain (g)	5.18 ± 1.34 ^b^	7.26 ± 1.41 ^a^	6.26 ± 1.40 ^b^	3.38 ± 1.11 ^c^
Kidney weight (g per 100 g BW)	1.41 ± 0.19	1.41 ± 0.11	1.37 ± 0.29	1.34 ± 0.15
Heart weight (g per 100 g BW)	0.64 ± 0.15 ^ab^	0.52 ± 0.07 ^b^	0.67 ± 0.13 ^a^	0.57 ± 0.12 ^ab^
Liver weight (g per 100 g BW)	3.27 ± 0.22 ^a^	3.26 ± 0.34 ^b^	3.18 ± 0.20 ^a^	3.65 ± 0.31 ^a^
Mesenteric WAT (g per 100 g BW)	0.47 ± 0.29 ^a^	2.52 ± 0.75 ^c^	1.81 ± 0.56 ^b^	0.66 ± 0.32 ^a^
Epididymal WAT (g per 100 g BW)	0.48 ± 0.16 ^a^	0.91 ± 0.16 ^c^	0.74 ± 0.16 ^b^	0.48 ± 0.12 ^a^
**Serum lipids**				
TG (mmol/L)	0.72 ± 0.14 ^b^	1.33 ± 0.30 ^a^	1.14 ± 0.23 ^a^	0.85 ± 0.21 ^b^
TC (mmol/L)	3.51 ± 0.31 ^c^	4.67 ± 0.42 ^a^	3.89 ± 0.36 ^b^	3.10 ± 0.22 ^d^
HDL-C (mmol/L)	2.13 ± 0.19 ^a^	2.08 ± 0.15 ^a^	2.19 ± 0.18 ^a^	2.06 ± 0.12 ^a^

Note: BW, body weight. WAT, white adipose tissue. The different lowercase letters indicate significant differences in the four groups (*p* < 0.05).

**Table 4 marinedrugs-21-00409-t004:** Effect of S-PLE and SC-PLE on liver lipids in MetS mice.

	TG (mmol/L)	TC (mmol/L)
ND	0.72 ± 0.14 ^b^	3.51 ± 0.31 ^c^
HFD	1.33 ± 0.30 ^a^	4.67 ± 0.42 ^a^
S-PLE	1.14 ± 0.23 ^a^	3.89 ± 0.36 ^b^
SC-PLE	0.85 ± 0.21 ^b^	3.10 ± 0.22 ^d^

Note: The different lowercase letters indicate significant differences in the four groups (*p* < 0.05).

**Table 5 marinedrugs-21-00409-t005:** Differential hepatic metabolites in ND vs. HFD.

Metabolite	Adduct	Class	VIP	Fold-Change	Type
Glutathione	[M-H]^−^	Amino acids and their metabolites	1.32	0.48	down
L-cystine	[M+H]^+^	Amino acids and their metabolites	1.51	2.86	up
Gamma-L-glutamic acid-l-cysteine	[M+H]^+^	Amino acids and their metabolites	1.47	0.32	down
Thromboxane B2	[M-H]^−^	Fatty acyl	1.45	2.23	up
Cortisol	[M+H]^+^	Hormones and hormone-related substances	1.93	0.31	down
Taurocholic acid	[M-H_2_O+H]^+^	Bile acid	1.37	0.2	down
13-oxo-9Z,11E-octadecenoic acid	[M-H]^−^	Fatty acyl	1.7	3.88	up
9-oxo-10E,12Z-octadecenoic acid	[M-H]^−^	Fatty acyl	1.7	3.88	up
9,12,13-trihydroxy-octadecenoic acid	[M-H]^−^	Fatty acyl	1.32	3.73	up
11-cis-retinol	[M+H]^+^	Coenzyme and vitamin	1.2	0.16	down
All-trans retinol	[M+H]^+^	Coenzyme and vitamin	2.11	0	down
Cysteinyl glycine	[M+H]^+^	Amino acids and their metabolites	1.51	0.35	down

**Table 6 marinedrugs-21-00409-t006:** Differential hepatic metabolites in HFD vs. S-PLE.

Metabolite	Adduct	Cless	VIP	Fold-Change	Type
N,N′-diacetyl chitobiose	[M+H]^+^	Carbohydrates and their metabolites	1.06	2.2	up
Carmine acid	[M+H]^+^	Amino acids and their metabolites	1.68	0.47	down
4-hydroxy-2-ketoglutaric acid	[M-H]^−^	Organic acids and their derivatives	1.14	0.47	down
DL-3,4-dihydroxy almond acid	[M-H]^−^	Organic acids and their derivatives	1.32	0.43	down
Deoxyguanosine diphosphate	[M-H]^−^	Nucleotide and metabolites thereof	1.37	2.06	up
5′-adenylate sulfuric acid	[M-H]^−^	Nucleotide and metabolites thereof	1.23	2.27	up
13-hydroxy-9Z,11E,15Z-octadecenoic acid	[M-H]^−^	Fatty acyl	1.43	0.49	down
Carnitine C8:0	[M+H]^+^	Fatty acyl	1.5	0.44	down
(±)17-HDHA	[M-H]^−^	Fatty acyl	2.17	3.97	up

**Table 7 marinedrugs-21-00409-t007:** Differential hepatic metabolites in HFD vs. SC-PLE.

Metabolite	Adduct	Class	VIP	Fold-Change	Type
Thromboxane B2	[M-H]^−^	Fatty acyl	1.24	0.41	down
Prostaglandin A2	[M-H]^−^	Fatty acyl	1.40	0.34	down
15-keto-prostaglandin F2a	[M-H]^−^	Fatty acyl	1.12	0.13	down
11-beta-prostaglandin PGF2a	[M-H]^−^	Fatty acyl	1.31	0.46	down
δ 12-prostaglandin J2	[M-H]^−^	Fatty acyl	1.40	0.34	down
reduced glutathione	[M-H]^−^	Amino acids and their metabolites	1.14	2.01	up
3,5-dihydroxy-3-methylpentanoic acid	[M-H]^−^	Organic acids and their derivatives	1.55	0.17	down
Gamma-L-glutamic acid-l-cysteine	[M+H]^+^	Amino acids and their metabolites	1.19	2.46	up
Cysteinyl glycine	[M+H]^+^	Amino acids and their metabolites	1.20	2.29	up
(±)5-HEPE	[M-H]^−^	Fatty acyl	1.84	6.07	up

## Data Availability

Not applicable.

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
