# Peer review of "Comparison of the Effect of Phospholipid Extracts from Salmon and Silver Carp Heads on High-Fat-Diet-Induced Metabolic Syndrome in C57BL/6J Mice"

_marinedrugs, 2023, doi:10.3390/md21070409_

Round 1

Reviewer 1 Report

In the paper entitled „Comparison of the effect of phospholipid extracts from salmon 2 and silver carp heads on high-fat diet-induced metabolic syn-3 drome in C57BL/6J mice“ authors studied the effect of salmon and silver carp phospholipid extracts on high-fat diet-induced metabolic syndrome by analysing Phospholipids Composition and metabolome in connection to expression of genes. After analysing all the data presented, I conclude that proposed manuscript is suitable for publication in Marine drugs with:

Major points:

- in Discussion section – the authors research the IRS/PI3K/AKT signalling pathway but in Discussion section this was completely omitted – please discuss obtained results

 - Conclusion is not clearly addressed because it should be a summary of the main topics covered by research – please add Conclusion as separate section

Minor points:

76…. IRS/PI3K/AKT signaling pathway ---- signalling.

83…. Signaling…..signalling

glyoxalate metabolism ………..glyoxylate metabolism

..check spelling

Author Response

We feel great thanks for your professional review work on our article. As you are concerned, there are several problems that need to be addressed. According to your nice suggestions, we have made extensive corrections to our previous draft, the detailed corrections are listed below. Please refer to the attachment for specific modifications.

Reviewer 2 Report

This study investigated the preventive effect of phospholipid extracts from salmon and silver carp heads on high fat diet-induced obesity and obesity-related metabolic disorders in C57BL mice. Overall, the results shown here suggest the metabolic improvements by S-PLE and SC-PLE. However, I have serious concerns that the discussion part have some problems with illogical description. Overall, it contains many misunderstandings and statements, I do not recommend this paper for the publication in this journal. I hope that my comment is useful for the improvement of the article.

1) Line 303-329, It is difficult to understand, the description needs to be re-written to remove ambiguities and confusion. It is better to discuss PUFA metabolism and the antioxidant activities of glutathione separately. I do not understand the premise the formation of beige adipocyte by ketoglutarate is related to the regulation of hepatic lipid metabolism by SC-PLE.

2) Line 330-340, Although the authors insist that the improving effect on metabolic disorders of S-PLE is superior to that of SC-PLE, the followed argument seems to contradict that claim. Moreover, the article by Gao et al.(43) is a paper showing the effect of EPA-PL and DHA-PL on intestinal dysfunction, but not on the regulatory effects on hepatic lipid metabolism. It is not appropriate to the description.

3) Line 129-137, I suggest adding the arrows in figure2 to help understand the reader for the discussion of the observation. Unfortunately, I do not think there are any changes such as the lipid droplets or cytoplasmic vacuolation the authors described.

4) Table 4, Is it meaningful to measure HDL-cholesterol in liver lipids? In the liver, cholesterol present as free cholesterol and cholesterol ester.

5) Line 269, What is the latex particles?

6) Line 295-302, What does “glucolipid metabolism” mean? It does not seem to be a misspelling of “glycolipids” either. The followed description also contains misunderstandings, and the reviewer cannot follow what the authors want to discuss.

Author Response

(The authors gave the same response as above.)

Round 2

Reviewer 1 Report

The Author's have made extensive corrections   to their previous draft and gave  answers  to all reviewer comments. 

But in line 151-152 Authors wrote an statement which is not in agreement to given  figure........... The levels of TNF-α, IL-6, MCP-1 and IL-1β in HFD group were significantly higher than those in the HFD group (P < 0.05).

Author Response

We feel great thanks for your professional review work on our article. The reviewer comments are laid out below in italicized font and specific concerns have been numbered. Our response is given in normal font and changes/additions to the manuscript are given in the red text. Please refer to the attachment for specific modifications.

Reviewer 2 Report

The manuscript has been much improved and is in a acceptable condition now,

Author Response

According to the reviewers’ comments, we have made extensive modifications to our manuscrip. Thank you again for your positive comments and valuable suggestions to improve the quality of our manuscript.